# Is Code Better Than Language for Algorithmic Reasoning?

**Terry Tong** [1]  **Yu Feng** [1]  **Surbhi Goel** [1]  **Dan Roth** [1]

## Abstract

For tool-augmented language models, comparing natural-language reasoning with code-execution pipelines is difficult because the comparison changes both the intermediate representation and the execution mechanism. We separate these factors with an intermediate intervention: the model expresses its reasoning as executable code, and the language model simulates that code in context to produce an answer. On a 40-task verifiable algorithmic benchmark, deterministic code execution outperforms natural-language reasoning by +31.6pp. We observe that the intermediate intervention is not meaningfully different from natural-language reasoning (+0.15pp). These results suggest that, in our evaluated setting, changing the intermediate representation alone does not explain the tool-use advantage, providing evidence for the performance gains requiring reliable external execution. We formalize this intuition with a simple statistical decision-theoretic model that characterizes when execution dominates end-to-end risk in our disentangled trace-generation/execution regime. We validate our theory using a reconstruction intervention that leverages a proxy language model to infer natural-language reasoning traces from code representations, recovering performance comparable to the original natural-language reasoning pipeline. All experiments are at https://github.com/TerryTong-Git/ToolProj.

## 1. Introduction

Many agentic systems orchestrate symbolic solvers, LLMs, and other tools, achieving state-of-the-art performance (Gao et al., 2023; Yao et al., 2023; Schick et al., 2023; Yang et al., 2024; Wang et al., 2025). Prior work shows that translating problems into solver-executable code (Route 3) and delegating execution often outperforms end-to-end NL

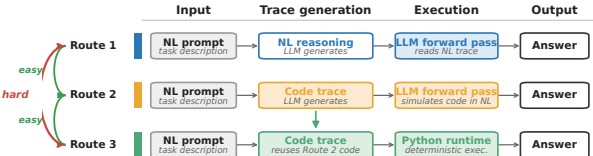

*Figure 1.* **Three-Route Framework.** We decompose algorithmic reasoning into: (1) *Translation* into NL or code, and (2) *Execution* via LLM or solver. This yields three routes: **Route 1** (Direct NL), **Route 2** (Code + NL simulation), **Route 3** (Code + Solver Execution). Prior work compares only Route 1 vs. Route 3, confounding translation and execution. Our Route 2 isolates these factors.

reasoning (Route 1) on logic- and algorithmic-style complex reasoning tasks (Lyu et al., 2023; Pan et al., 2023).

It is unclear whether these gains come from the structured code representation, the reliability of an external executor, or both. A direct comparison is ill-posed because the routes learn different objects: natural-language traces versus solver-executable programs. Without a common intermediate route, performance gaps conflate representation and execution.

This paper presents a systematic three-route framework (Figure 1 and Section 2) that makes the comparison tractable. We decouple the pipeline into trace generation (e.g. Chain of Thought (Wei et al., 2022)) and execution (see Section 2.2), instantiated by controlled prompts in Figure 2. Route 2 fixes the executor class while changing the representation through code generation followed by LLM simulation. Route 3 reuses the same code trace and delegates execution to Python.

Empirically, the framework yields Route 1 (NL + NL reasoning) $\approx$ Route 2 (code + NL simulation) < Route 3 (code + Python execution) (Section 3.2). On the 40 task benchmark, the accuracies are 17.21%, 17.37%, and 48.84%, respectively. The paired Route 2–Route 1 gap is +0.15pp with 95% cluster-bootstrap CI $[-0.30, +0.61]$pp. Route 3 exceeds Route 2 by +31.47pp with CI $[+29.20, +33.71]$pp (Figures 3 and 4).

To test representation, we hold the LLM executor fixed and vary only the trace. Theory shows that code is risk non-inferior when natural language adds nuisance variation. The nuisance assumption seems to align with our intuition that a problem solution can be expressed in more ways in natural language than in code. A reconstruction experiment empirically demonstrates that code-derived NL traces also recover native-NL performance (Section 4.3 and Figure 5). Thus, code retains decision-relevant information, and representa-

*Equal contribution [1]University of Pennsylvania. Correspondence to: Terry Tong <tongt1@seas.upenn.edu>.

*Proceedings of the 43rd International Conference on Machine Learning*, Seoul, South Korea. PMLR 306, 2026. Copyright 2026 by the author(s).

*Figure 2.* **Prompt templates for three-route evaluation. Route 1 (NL)**: LLM reasons in natural language only, code forbidden. **Route 2 (Sim)**: LLM generates Python `solution()` then simulates execution in NL. **Route 3 (Code)**: Same code executed in Python runtime. This isolates execution mechanism while controlling translation.

tion is not the primary bottleneck in this setting.

We then analyze Route 2 vs. Route 3 (Section 5), where the code trace is fixed and only the executor changes. The recovery mass where Route 2 succeeds while Route 3 fails is 1.61%, and the execution-win mass where Route 3 succeeds while Route 2 fails is 33.08% (Figure 6). As task difficulty increases, code execution maintains substantially higher accuracy. We therefore attribute the gap in our evaluated setting to reliable execution rather than code trace generation.

Our main contributions are:

1. A **three-route framework** (Section 2 and Figures 1 and 2) for tractable comparison between code and natural language representations via an intermediary (code generation with LLM execution).
2. **Empirical validation** (Section 3 and Figures 3 and 4) demonstrating that Route 1 and Route 2 are statistically close, while Route 3 is substantially better.
3. A **systematic study** (Sections 4 and 5) ruling out trace generation as the bottleneck and solidifying execution as the bottleneck in end-to-end algorithmic reasoning performance when using language.

## 2. Three-Route Framework

Our central research question (RQ) is: *Is code > NL for algorithmic reasoning?* To begin to answer this question, we first introduce our three-route framework which enables tractable comparison by disentangling *reasoning representation* (hereafter called Traces) and *reasoning execution* (hereafter called Executors) and constructing an intermediary bridge (Route 2) for pairwise comparison. This section introduces the task (Section 2.1), notation (Section 2.2), and route definitions (Section 2.3).

### 2.1. Task and Loss

For a gold evaluation distribution over task instances $i$ specified by (algorithm, input variables, seed), let $X \sim p(x)$ denote a task instance (problem statement and inputs), and let $Y^*(X) \in \mathcal{Y}$ denote the ground-truth output that is unique, fixed, and externally verifiable. We evaluate performance under 0–1 loss:

$$\ell(y, x) := \mathbf{1}\{y \neq Y^*(x)\}.$$

### 2.2. Trace Generators and Executors

We model each reasoning pipeline as a two-stage stochastic process with *(1) trace generation* and *(2) execution*.

**Trace generation.** We define a *trace* as an object that stores intermediate reasoning used to solve a corresponding task (e.g. NL Chain-of-Thought or a program). A *trace generator* is a (stochastic) mapping

$$E : \mathcal{X} \rightarrow \Delta(\mathcal{Z}), \qquad Z \sim p_E(z \mid x),$$

which produces an auxiliary trace $Z$ given the task instance.

**Execution.** We define *execution* as a procedure that consumes a trace, e.g. an LLM forward pass that takes as input a reasoning trace and outputs an answer, or executing a program in an external runtime. An *executor* is a (stochastic) mapping

$$\rho : \mathcal{X} \times \mathcal{Z} \rightarrow \Delta(\mathcal{Y}),$$

mapping the observed instance and trace to a final output.

### 2.3. The Three Routes

We consider three reasoning pipelines ("routes"), illustrated in Figure 1. Each route is represented as a pair $(E, \rho)$ consisting of a trace generator $E$ and an executor $\rho$.

**Route 1 (Direct Natural Language).** Route 1 represents standard NL reasoning: the model first produces a natural-language (Chain-of-thought) trace and then the same model is used as the LLM-based executor, conditioning on the trace to generate a final answer. Formally, a natural-language trace generator $E_{\mathrm{NL}}$ produces traces $Z_{\mathrm{NL}} \sim p_{\mathrm{NL}}(\cdot \mid X)$, paired with an executor family

$$\mathcal{H}_{\mathrm{NL}} \subseteq \{\rho : \mathcal{X} \times \mathcal{Z}_{\mathrm{NL}} \rightarrow \Delta(\mathcal{Y})\}.$$

**Route 2 (Code + NL Simulation).** Route 2 uses *code* as the trace modality, but keeps execution "in-model": the LLM instead simulates the generated code in natural language, rather than executing it in an external environment. This route isolates the effect of using a code trace while holding the LLM-based executor class fixed. Formally, a code trace generator $E_{\mathrm{Code}}$ produces executable representations $Z_{\mathrm{C}} \sim p_{\mathrm{Code}}(\cdot \mid X)$, paired with an executor family

$$\mathcal{H}_{\mathrm{C}} \subseteq \{\rho : \mathcal{X} \times \mathcal{Z}_{\mathrm{C}} \rightarrow \Delta(\mathcal{Y})\}$$

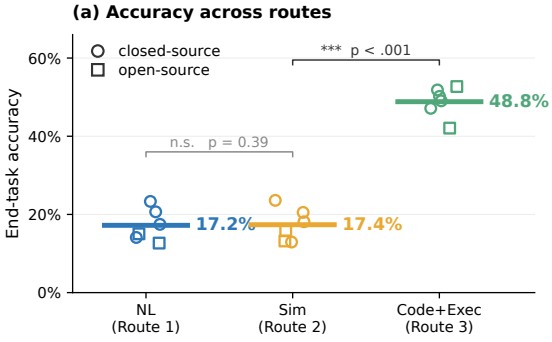
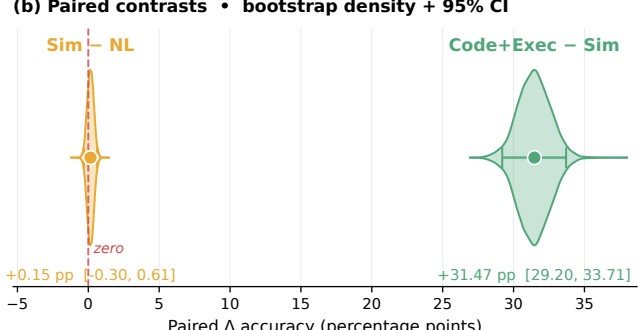

Figure 3. **Code + Solver Execution performs better than Direct NL reasoning quantified by end-task accuracy overall and within paired instances.** Paired instance contrasts are shown in the bootstrap distributions. Results are averaged over the 40-task analysis set with 1,113 unique problem instances, 3 seeds, and 6 models.

corresponding to language-model-based simulation of code execution.

**Route 3 (Code + Solver Execution).** Route 3 uses the same code trace generator $E_{\text{Code}}$ as Route 2, producing $Z_{\text{C}} \sim p_{\text{Code}}(\cdot \mid X)$, but pairs it with a deterministic executor corresponding to external code execution. Let $\text{Exec} : \mathcal{X} \times \mathcal{Z}_{\text{C}} \to \mathcal{Y}$ denote the (fixed) external runtime that executes the code trace $z$ on instance $x$ and returns an output. Our executor is

$$\rho_{\text{Exec}}(y \mid x, z) := \mathbf{1}\{y = \text{Exec}(x, z)\}.$$

The corresponding executor family is the singleton $\mathcal{H}_{\text{Exec}} := \{\rho_{\text{Exec}}\}$.

## 3. Evaluating the Three-Route Framework

Our evaluation aims to answer the research questions (RQ): 1) Does Route 1 $\approx$ Route 2 $<$ Route 3 hold? (Sections 3.1 and 3.2)

### 3.1. Experimental Instantiation of the Three Routes

To draw conclusions about trace modality and execution method, we fix one stage at a time. For Route 1 vs. Route 2, we fix the execution phase by using the same LLM reasoning model forward pass, but use different modalities in the trace generator. For Route 2 vs. Route 3, we fix the trace generator that outputs code, then use different execution methods: either an LLM reasoning model forward pass or a Python 3 runtime. Below, let $X_i$ be the task instantiation of instance $i$. Structured JSON outputs are enforced for sampling in all routes (Figure 2).

**Sampling Route 1.** Route 1 prompts an LLM $E_{NL}$ to function as a trace generator $Z_{NL} := E_{NL}(X_i)$. The trace is then fed into the same LLM $\rho_{NL} = E_{NL}$ to produce an answer $Y_i^{(NL)} := \rho_{NL}(Z_{NL})$. The prompt instructs the model to never use code, and to output a structured rationale and answer (see Figure 2). Operationalized, one experimental run for Route 1 is encapsulated in a single LLM reasoning model's forward pass without pause.

**Sampling Route 2.** Similarly, Route 2 uses a LLM $E_{Code}$ as a trace generator for code modality $Z_{Code} := E_{Code}(X_i)$. The trace is then fed into the same LLM $\rho_{Sim} = E_{Code}$ to produce an answer $Y_i^{(Sim)} := \rho_{Sim}(Z_{Code})$. Operationalized, one experimental run for Route 2 is encapsulated in a single LLM reasoning model's forward pass without pause. Prompt templates are controlled to be as similar as possible, with Route 2 differing only by the inclusion of a code-generation and simulation instruction (Figure 2). We prompt the model to output a program snippet, structured reasoning over the code, followed by an answer.

**Sampling Route 3.** Route 3 takes the exact same code trace $Z_{Code}$ as Route 2, except it runs it through a Python 3 runtime $\rho_{Exec}$ and retrieves the solution $Y_i^{(Exec)} = \rho_{Exec}(Z_{Code})$. The Python 3 runtime has access to five standard scientific libraries: `SciPy`, `NumPy`, `pandas`, `PuLP`, and `PyTorch`.

**Observed outcomes.** For each problem instance $i$, we observe paired Bernoulli outcomes

$$\left( Y_i^{(\text{NL})}, \; Y_i^{(\text{Sim})}, \; Y_i^{(\text{Exec})} \right).$$

**Data and models.** We evaluate a 40-task analysis set drawn from CLRS-30, NP-Hard-Eval, and a custom fine-grained evaluation suite, across three random seeds $\{0, 1, 2\}$. The set contains 1,113 unique problem instances and 20,034 paired route evaluations after pooling six full-coverage models. Tasks span arithmetic, dynamic programming, graph algorithms, string algorithms, geometry, sorting, and NP-hard optimization, with difficulty controlled by a parameter $\tau$ when applicable. We evaluate closed-source models (`Claude Haiku 4.5`, `GPT-4o-mini`, `Gemini 2.0 Flash`, `Gemini 2.5 Flash`) and open-source models (`Mixtral-8x22b-Instruct`, `Codestral-2508`). Appendices A.1 and A.2 stratify these results by complexity and model, and Appendix C reports subset model coverage.

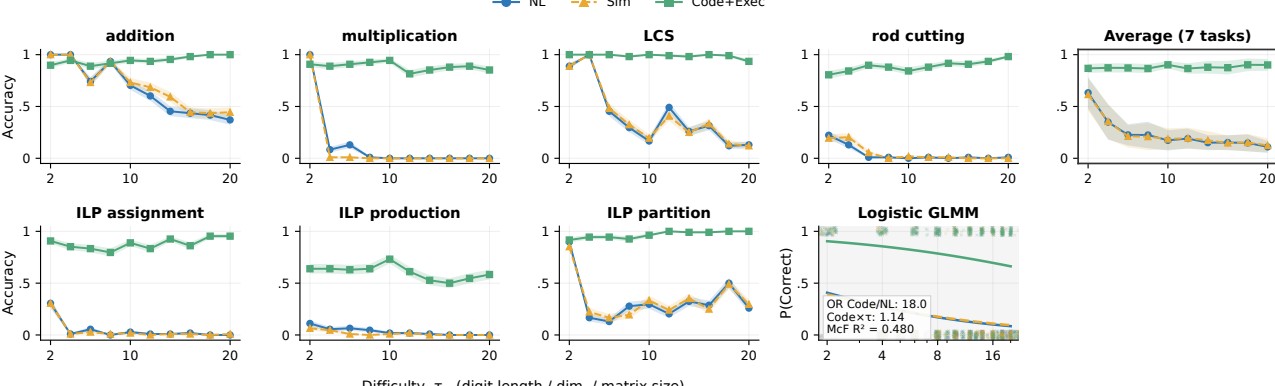

*Figure 4.* **Code + Solver Execution scales better than natural language reasoning when problems get harder, both within-task and on average, interpolating and extrapolating (gray).** $\tau$ is used to control digit length for arithmetic, table dimensionality for dynamic programming, and constraint matrix dimensionality for integer linear programming. The figure uses the same data and setup as Figure 3.

## 3.2. End-to-End Performance Comparison

We first define the paired tests used to compare the three routes, then report the pooled route accuracies and paired differences, and finally summarize how the gaps change with task difficulty.

**Pairwise statistical tests.** We evaluate pairwise route differences using McNemar tests on paired Bernoulli outcomes. For Route 1 vs. Route 2, the null hypothesis is

$$H_0: \ \Pr(Y^{(\mathrm{NL})} = 1, Y^{(\mathrm{Sim})} = 0)$$
$$= \Pr(Y^{(\mathrm{NL})} = 0, Y^{(\mathrm{Sim})} = 1),$$

with an analogous null for Route 2 vs. Route 3.
Effect sizes are reported as paired accuracy differences, e.g.,

$$\Delta_{\mathrm{Sim-NL}} = \mathrm{Acc}(\mathrm{Sim}) - \mathrm{Acc}(\mathrm{NL}),$$

with 95% confidence intervals estimated via cluster bootstrap resampling over instances. Holm–Bonferroni correction is applied to control family-wise error rate of the above 2 marginal pairwise tests on fully pooled data at $\alpha = 5\%$.

**Difficulty scaling.** To analyze how performance varies with task difficulty, we fit a generalized linear mixed-effects model (GLMM) for binary accuracy $Y_i \in \{0, 1\}$ with a logistic link:

$$Y_i \mid u_{\mathrm{inst}[i]}, u_{\mathrm{seed}[i]} \sim \mathrm{Bernoulli}(p_i),$$

$$\mathrm{logit}(p_i) = \alpha + \beta_{\mathrm{route}_i} + \gamma\,\tau_i + \delta_{\mathrm{route}_i}\,\tau_i + u_{\mathrm{inst}[i]} + u_{\mathrm{seed}[i]},$$

where $u_{\mathrm{inst}[i]} \sim \mathcal{N}(0, \sigma_{\mathrm{inst}}^2)$ and $u_{\mathrm{seed}[i]} \sim \mathcal{N}(0, \sigma_{\mathrm{seed}}^2)$ are random intercepts. Route and difficulty are modeled as fixed effects (including a route $\times$ difficulty interaction), with instance and seed modeled as random effects.

**Results.** Across the 40-task analysis set (Figure 3), Route 1 reaches 17.21% accuracy, Route 2 reaches 17.37%, and Route 3 reaches 48.84%. *Route 3 has a statistically significant advantage over Route 2*, with a +31.47pp

paired accuracy gap and a 95% cluster-bootstrap interval of $[+29.20, +33.71]$pp. For Route 1 vs. Route 2, the paired gap is +0.15pp with a 95% cluster-bootstrap interval of $[-0.30, +0.61]$pp and a two-sided McNemar $p = 0.39$, so we do not conclude a meaningful difference between natural-language reasoning and code simulation under this evaluation. On the other hand, as performance gaps widen with increasing task difficulty (Figure 4), Route 3 remains substantially higher as the language-based routes degrade. This motivates the hypothesis that execution is the bottleneck.

## 4. Route 1 vs. Route 2 Analysis

In Section 3.2, Route 1 and Route 2 achieved similar paired accuracies under our evaluated models and prompts. This section asks: **Can we provide evidence that input representation is not the bottleneck to end-task performance?** We first model the theory in the simple linear case (Sections 4.1 and 4.2), then test the corresponding language-interface prediction (Section 4.3).

The main result is that when we model code distribution as a canonicalization of natural language distribution, we can show for our simple linear hypothesis class that the uniform worst-case risk over *all* hypotheses is exactly 0. In other words, under these conditions, "code" reasoning is non-inferior to "natural language" reasoning.

### 4.1. Linear Modeling Setup and Intuition

Controlling the downstream hypothesis class enables us to compare the effect of the two input representations $P_C$ and $P_N$, both of which live in the same feature space. For convenience, the downstream hypothesis class is modeled as realizable and linear.

$$\mathcal{H}_{\mathrm{lin}} = \left\{ h_{a,v}(b, u) = a^\top b + v^\top u : a \in \mathbb{R}^{d_B},\ v \in \mathbb{R}^{d_U} \right\}. \tag{1}$$

Let $r \in \{C, N\}$, where $C$ denotes code and $N$ denotes native natural language. For each task, route $r$ produces a

random input:

$$S_r = (B, U_r) \sim P_r, \qquad S_r \in \mathbb{R}^{d_B + d_U}. \qquad (2)$$

The vector $B \in \mathbb{R}^{d_B}$ is the answer-relevant core, or the sufficient statistic (Lehmann & Casella, 1998). The vector $U_r \in \mathbb{R}^{d_U}$ contains route-specific surface variation that should not change the answer once the core is fixed. Prior research in NLP / LLMs has shown that models are sensitive to syntactic heuristics, prompting, lexical choice, etc (Gururangan et al., 2018; McCoy et al., 2019; Geirhos et al., 2020; Zhao et al., 2021; Lu et al., 2022; Pezeshkpour & Hruschka, 2023).

As a running example, consider a dynamic-programming problem. The core $B$ contains the recurrence, boundary conditions, and final query that determine the answer. A code trace may still vary in decision-irrelevant details such as variable names, helper-function names, or formatting; these are part of $U_C$. A natural-language trace can express the same recurrence, but it also admits extra paraphrases such as "fill the table from smaller subproblems," "reuse previous states," or "work backward from the target." These additional surface choices are the extra term $\eta_N$. If both routes preserve $B$, then those prose choices should not create new algorithmic information.

We define the linear-model assumption here before proving the risk comparison. Following the intuition presented above, we state the assumption as a shared-core generative model with extra natural-language nuisance. The first-order orthogonality conditions imply the centering and risk-separation facts defined in the lemma.

**Assumption 4.1.** There exist variables $B$, $U_C$, $\eta_N$, and $\varepsilon$ such that

$$S_C = (B, U_C), \qquad S_N = (B, U_N), \qquad U_N = U_C + \eta_N, \qquad (3)$$

and

$$Y = \theta^\top B + \varepsilon. \qquad (4)$$

The code-side nuisance is centered conditional on the core, the extra natural-language nuisance is mean-zero conditional on the core and code-side nuisance, and the residual label noise is mean-zero conditional on all nuisance coordinates:

$$\begin{aligned} \mathbb{E}[U_C \mid B] &= 0, \\ \mathbb{E}[\eta_N \mid B, U_C] &= 0, \qquad (5) \\ \mathbb{E}[\varepsilon \mid B, U_C, \eta_N] &= 0. \end{aligned}$$

The residual nuisance $\eta_N$ represents additional paraphrase, verbosity, ordering, style, and prompt-form variation in native natural language. The direction $U_N = U_C + \eta_N$ formalizes the claim that natural language carries the code-side nuisance plus additional surface variation. Code has a more constrained grammar and conventional structure, whereas natural-language explanations admit more paraphrastic and discourse-level variation. Work on the naturalness of software likewise treats code as a structured statistical object with regularities distinct from ordinary natural language (Allamanis et al., 2018).

The next lemma is necessary and load-bearing for the main theorem. The lemma states that, once the core is fixed, both routes' surface coordinates remain centered nuisance and do not secretly carry residual label information.

**Lemma 4.1.1.** *Under Assumption 4.1, for $r \in \{C, N\}$,*

$$\mathbb{E}[U_r \mid B] = 0, \qquad (6)$$

$$\mathbb{E}[\varepsilon \mid B, U_r] = 0, \qquad (7)$$

*and*

$$\mathbb{E}[BU_r^\top] = 0, \qquad \mathbb{E}[\varepsilon U_r] = 0. \qquad (8)$$

*Moreover, the first-order nuisance condition implies the cross-moment identity:*

$$\mathbb{E}[U_C \eta_N^\top \mid B] = 0. \qquad (9)$$

*Proof.* For code, $\mathbb{E}[U_C \mid B] = 0$ is part of Assumption 4.1. For natural language,

$$\begin{aligned} \mathbb{E}[U_N \mid B] &= \mathbb{E}[U_C \mid B] + \mathbb{E}[\eta_N \mid B] \\ &= 0 + \mathbb{E}[\mathbb{E}[\eta_N \mid B, U_C] \mid B] \\ &= 0. \end{aligned}$$

The label-noise condition follows by the tower property because $U_C$ and $U_N = U_C + \eta_N$ are both functions of $(B, U_C, \eta_N)$. Thus $\mathbb{E}[\varepsilon \mid B, U_r] = 0$ for $r \in \{C, N\}$. The two risk-separation equalities follow from

$$\mathbb{E}[BU_r^\top] = \mathbb{E}[B \, \mathbb{E}[U_r^\top \mid B]]$$

and

$$\mathbb{E}[\varepsilon U_r] = \mathbb{E}[U_r \, \mathbb{E}[\varepsilon \mid B, U_r]] = 0.$$

Finally,

$$\mathbb{E}[U_C \eta_N^\top \mid B] = \mathbb{E}[U_C \, \mathbb{E}[\eta_N^\top \mid B, U_C] \mid B] = 0.$$

$\square$

Intuitively, the lemma says that the extra natural-language words are not allowed to be a hidden answer channel once the algorithmic core is known.

## 4.2. Results

We now make the representation claim precise in the linear model. The argument has three steps: define a nuisance covariance ordering, show that the extra-native-language-nuisance assumption implies that ordering, and then use the ordering to compare the risk of every member of the linear hypothesis class under the two route distributions. We end with intuition that leads to the experimental results.

Let

$$\Sigma_{U,r} = \mathbb{E}[U_r U_r^\top] \qquad (10)$$

be the route-$r$ nuisance covariance. We use the Loewner order on positive semidefinite matrices: $A \preceq B$ means $B - A$ is positive semidefinite, equivalently $v^\top A v \leq v^\top B v$ for every vector $v$.

The following proposition formalizes the following intuition. If native natural language is the code-side nuisance plus extra paraphrase, then the natural-language nuisance covariance cloud should be at least as spread out as the code nuisance cloud in every linear direction. In other words, conditional on the same core, the additional natural-language degrees of freedom do not supply extra answer signal.

**Proposition 4.2.** *Under Assumption 4.1,*

$$\Sigma_{U,C} \preceq \Sigma_{U,N}. \qquad (11)$$

*Proof.* Using (3),

$$\Sigma_{U,N} = \mathbb{E}[(U_C + \eta_N)(U_C + \eta_N)^\top] \qquad (12)$$
$$= \mathbb{E}[U_C U_C^\top] + \mathbb{E}[\eta_N \eta_N^\top] + \mathbb{E}[U_C \eta_N^\top] + \mathbb{E}[\eta_N U_C^\top]. \qquad (13)$$

The cross terms vanish by Lemma 4.1.1:

$$\mathbb{E}[U_C \eta_N^\top] = \mathbb{E}\big[\mathbb{E}[U_C \eta_N^\top \mid B]\big] = 0, \qquad (14)$$

and similarly for its transpose. Hence

$$\Sigma_{U,N} - \Sigma_{U,C} = \mathbb{E}[\eta_N \eta_N^\top] \succeq 0. \qquad (15)$$

$\square$

The covariance order from the above proposition implies the uniform population risk non-inferiority for code representations.

**Theorem 4.3.** *Under Assumption 4.1, for squared loss and the common class $\mathcal{H}_{\text{lin}}$ in (1),*

$$\sup_{h \in \mathcal{H}_{\text{lin}}} \{R_C(h) - R_N(h)\} \leq 0. \qquad (16)$$

*Consequently code is representation-non-inferior to native natural language with margin 0 in this linear model.*

*Proof.* Fix $h_{a,v} \in \mathcal{H}_{\text{lin}}$. By (4),

$$h_{a,v}(B, U_r) - Y = (a - \theta)^\top B + v^\top U_r - \varepsilon. \qquad (17)$$

Using Lemma 4.1.1, the terms involving $U_r$ separate:

$$R_r(h_{a,v}) = R_{\text{core}}(a) + v^\top \Sigma_{U,r} v, \qquad (18)$$

where $R_{\text{core}}(a) = \mathbb{E}[((a - \theta)^\top B - \varepsilon)^2]$ does not depend on $r$. Therefore

$$R_C(h_{a,v}) - R_N(h_{a,v}) = v^\top (\Sigma_{U,C} - \Sigma_{U,N}) v \leq 0 \quad (19)$$

by (11). Taking the supremum over $h \in \mathcal{H}_{\text{lin}}$ proves (16).

$\square$

Since the core term is identical for code and native natural language, the only possible representation-level difference is the penalty for depending on $U_r$. Because native language has no smaller nuisance covariance, it cannot lower that penalty. Thus, in this model, trace representation in Route 1 cannot be the bottleneck for improving performance, since the core algorithmic information is already present in the code representation [1].

*Remark* 4.4 (Finite-sample OLS intuition). By studying why nuisance coordinates hurt learning in the finite-sample regime, we can gain intuition about the uniform population risk result above[2]. Let $p = d_B + d_U$ and consider the Gaussian special case

$$S_r \sim \mathcal{N}(0, \Sigma_r), \qquad Y = \beta^{\star\top} S_r + \varepsilon, \qquad \beta^\star = (\theta, 0), \qquad (20)$$

with $\varepsilon \sim \mathcal{N}(0, \sigma^2)$. Ordinary least squares and ridge regression have finite-sample excess-risk terms controlled by dimension, conditioning, and noise variance. In the isotropic case $\Sigma_r = I_p$, if $\widehat{\beta}_r$ is ordinary least squares from $n > p+1$ independent route-$r$ samples, then

$$\mathbb{E}_{D_{r,n}}[R_r(\widehat{\beta}_r)] - R^\star = \sigma^2 \frac{p}{n - p - 1}, \qquad (21)$$

where $R^\star = \sigma^2$. Fitting irrelevant coordinates increases finite-sample prediction error (Hastie et al., 2009; Wainwright, 2019). In the running example, this is the difference between learning from the recurrence itself and learning from many alternative descriptions of the same recurrence. Extra wording can be harmless in the infinite-data idealization. However, it is not a new source of answer-relevant information and can be a nuisance channel that a finite model must learn not to use.

Overall, the linear theory provides an intuition for why representation is not the bottleneck. Once both routes preserve the same core, extra native-language nuisance can only increase population risk for this hypothesis class. Section 4.3 demonstrates the nuisance intuition empirically when the theory breaks down.

### 4.3. Decision Intervention Reduction

When the theory breaks down due to non-linearity and high-dimensionality, we show that the practical implications remain. If translating code into natural language via fixed transformation, similar to our nuisance intuition, preserves downstream accuracy relative to native natural-language reasoning, then the representation non-inferiority of code remains the same even outside the linear proof.

---

[1] If this were not the case, we would not be able to get the final answer output from our pipeline.

[2] In the uniform case, we do not select a hypothesis in the hypothesis class like in the learning case, but rather, we show the phenomenon is true across *all* hypotheses in the class.

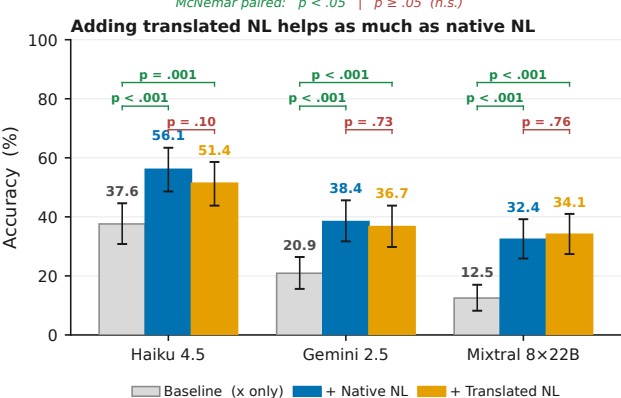

*Figure 5.* **Translated NL has similar downstream functionality to native NL as measured by end-task accuracy.** When we concatenate reasoning traces to the prompt, accuracy rises above the question-only baseline, indicating that the model uses the reasoning. Baseline-vs-concatenated differences are statistically significant, while Translated NL and Native NL are not significantly different. Source data is randomly sampled from the data used to generate Figure 3; we use 1000 samples per model.

**Intervention setup.** Our goal is to provide evidence that code is non-inferior to NL in expected loss in the LLM regime. In our proof, we modeled NL as a fixed transformation (linearly additive projection) of a code latent. Similarly, we apply a fixed transformation of the code (e.g., permutations and syntactic shifts) to simulate NL. We expect NL to yield nothing worse, since the underlying assumption is that code contains all decision-relevant information. The following experiment tests this implication directly.

We show that a fixed transformation of code (to natural language) behaves functionally similar to original natural-language reasoning. Here, our estimand is final accuracy, and the treatment is one of three inputs to the LLM executor: translated natural-language reasoning, original natural-language reasoning, or the baseline prompt without reasoning.

For a task instance $x$, we prompt a target language model in three conditions: (1) Baseline: x, (2) Native: $x \parallel z_{\mathrm{NL}}$, (3) Translated: x $\parallel \hat{z}_{\mathrm{NL}}$, where $z_{\mathrm{NL}} \sim p_{\mathrm{NL}}(\cdot \mid x)$ is the native Route 1 trace and $\hat{z}_{\mathrm{NL}}$ is obtained by translating the corresponding code to NL using the same translator procedure. If translated NL loses decision-relevant information relative to native NL, we would expect

$$\mathrm{Acc}(x \parallel \hat{z}_{\mathrm{NL}}) < \mathrm{Acc}(x \parallel z_{\mathrm{NL}}).$$

**Protocol and models.** We run this test on 1000 held-out instances across multiple translator models (`Claude-Haiku-4.5`, `Gemini-2.5-Flash`, `Mixtral-8x22B-Instruct`). Crucially, in this experiment the translator model is matched to the original code generator (i.e., we translate code produced by the same model family), to avoid confounding functional losses with cross-model stylistic mismatch.

**Results.** We fail to reject the null hypothesis of equal performance and observe overlapping 95% confidence intervals between the Native and Translated conditions (Figure 5). This suggests that, under our evaluated settings, translating code into NL does not destroy decision-relevant information for downstream answering, consistent with the claim that NL CoT does not systematically add algorithmic advantage beyond what is in code. Qualitatively, the translated results are similar to the originals, though this appears to depend heavily on the translating prompt.

**Interpretation.** Our empirical results indicate that translated code traces and native natural-language traces provide similar decision-relevant information in the settings we study. We find no evidence that natural-language reasoning introduces novel algorithmic strategies beyond those already captured by code representations on algorithmic tasks. Thus, in these experiments, replacing NL traces with code traces in the generation stage is not the end-to-end bottleneck. The execution analysis in Section 5 examines the remaining gap.

## 5. Route 2 vs. Route 3 Analysis

The key research question in this section is: **Is execution the bottleneck for language-based reasoning?** To address this question, we also ask two sub-questions:
1) When code is *correct*, does external code execution (Route 3) have lower expected loss than LLM-based code execution (Route 2)?
2) When code is *incorrect*, is the probability that Route 2 outperforms Route 3 small?

We pinpoint execution as the bottleneck (cf. Section 4), highlighting how using LLMs to generate code plans and then delegating to an external solver is the best-performing of the three evaluated routes (Section 3.2). In the 40-task analysis set, Route 3 exceeds Route 2 by +31.47pp. The recovery mass where Route 2 succeeds while Route 3 fails is 1.61%, whereas the execution-win mass where Route 3 succeeds while Route 2 fails is 33.08%.

### 5.1. Setup

As defined in Section 2, Route 2 and Route 3 share the same trace generator $E_{\mathrm{Code}}$ and differ only in the executor.

Let $\rho_{\mathrm{Sim}} \in \mathcal{H}_{\mathrm{C}}$ denote the fixed LLM-based simulation executor used in Route 2. Let $g$ be a deterministic interpreter mapping $(x, z_{\mathrm{C}})$ to $\mathcal{Y} \cup \{\perp\}$, where $\perp$ denotes execution failure. Define the instance-correct execution event

$$C := \{g(X, Z_{\mathrm{C}}) = Y^*(X)\}.$$

**Risk decomposition.** Let

$$e_C := \Pr(\hat{Y}_{\mathrm{Sim}} \neq Y^*(X) \mid C),$$
$$r := \Pr(\hat{Y}_{\mathrm{Sim}} = Y^*(X) \mid \neg C).$$

Then

$$R(E_{\text{Code}}, \rho_{\text{Exec}}) = \Pr(\neg C),$$
$$R(E_{\text{Code}}, \rho_{\text{Sim}}) = \Pr(C)\, e_C + \Pr(\neg C)(1 - r),$$

and hence

$$R(E_{\text{Code}}, \rho_{\text{Sim}}) - R(E_{\text{Code}}, \rho_{\text{Exec}})$$
$$= \Pr(C)\, e_C - \Pr(\neg C)\, r.$$

**Implications.** Simulation noise on correct-execution instances ($\Pr(C)e_C$) harms Route 2, while recovery on incorrect or failed executions ($\Pr(\neg C)r$) helps Route 2. Route 3 dominates whenever the recovery mass is insufficient to offset simulation noise, a condition quantified empirically in Section 3.2.

**Empirical recovery as an upper bound.** A direct way to operationalize the "recovery" term in the decomposition is to measure the frequency with which Route 2 succeeds while Route 3 fails on the *same* generated code trace, i.e.,

$$\Pr\!\left(\hat{Y}_{\text{Sim}} = Y^*(X),\ g(X, Z_{\text{C}}) \neq Y^*(X)\right).$$

This quantity corresponds to the *recovery mass* $\Pr(\neg C)\, r$ appearing in the above risk gap. It aggregates both (i) genuine recovery from incorrect code and (ii) cases where execution fails (e.g., $g(X, Z_{\text{C}}) = \bot$) but the simulator still answers correctly. Because the simulator may partially ignore $Z_{\text{C}}$ and answer directly from $X$, this statistic should be interpreted as an *upper bound* on mechanistic "recovery from flawed code."

**Interpretation.** Route 2 can outperform Route 3 only if the recovery mass $\Pr(\neg C)\, r$ is large enough to offset simulation noise on correct-execution instances $\Pr(C)\, e_C$. Empirically, we find that recovery mass is consistently small across tasks and models (Figure 6), indicating that Route 2 rarely compensates for execution failures[3] via recovery. In the 40-task analysis set, this recovery mass is 1.61%, and the execution-win mass $\Pr(C)e_C$ is 33.08%. These values explain why deterministic execution typically achieves lower end-to-end error than simulation on the same generated code traces (Figure 3).

# 6. Related Work and Discussion

**Neuro-symbolic Learning.** This paper builds on research in neuro-symbolic integration (Graves et al., 2014; Veličković & Blundell, 2021; Reed & Freitas, 2016; Graves et al., 2016), which combines neural networks with symbolic reasoning systems. These approaches are motivated by cognitive science (Schneider & Chein, 2003; Risko & Gilbert, 2016; Anderson, 2010), hierarchical reinforcement learning (Kolter

---

[3]Table 7 reports the corresponding code-execution failure modes.

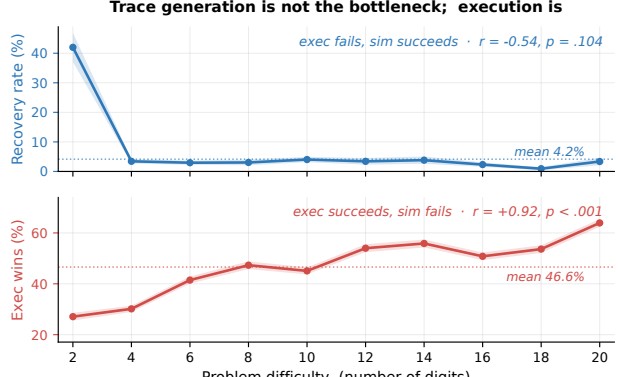

**Trace generation is not the bottleneck; execution is**

*Figure 6.* **Recovery mass (blue) is 1.61% overall, while execution-win mass (red) is 33.08% overall.** Recovery mass $\Pr(\neg C)\, r$ counts cases where code execution fails but LLM simulation succeeds. Execution-win mass $\Pr(C)e_C$ counts cases where code execution succeeds and LLM simulation fails.

et al., 2007; Dieterich, 2000), and compositionality research (Hudson & Manning, 2018; Hupkes et al., 2020; Andreas et al., 2017; Poggio et al., 2017). An orthogonal line of work explores direct execution of algorithms by neural networks (Veličković & Blundell, 2021; Mahdavi et al., 2023; Ibarz et al., 2022; Yan et al., 2020). Unlike these approaches that focus on *how* to integrate neural and symbolic components, our work addresses *why* neuro-symbolic integration outperforms neural reasoning alone for algorithmic tasks (Sections 4 and 5).

**LLM Reasoning.** Recent work has explored various reasoning paradigms for LLMs, including symbolic reasoning (Marra et al., 2019; Olausson et al., 2023; Han et al., 2024), chain-of-thought prompting (Wei et al., 2022; Zelikman et al., 2022; Merrill & Sabharwal, 2024; Altabaa et al., 2025), and in-context learning (Xie et al., 2021; Garg et al., 2022; Akyürek et al., 2022; Zhang et al., 2024). Xie et al. (2021) model in-context learning as implicit Bayesian inference, which we extend to compare different reasoning representations. While prior work demonstrates *that* certain prompting strategies improve performance, we provide a theoretical framework (Section 2) explaining *why* code representations are never worse than natural language (Section 4) in certain settings.

**LLM Tool-Use.** Tool-augmented LLMs have achieved strong empirical results (Shen, 2024; Schick et al., 2023; Qin et al., 2023; Tang et al., 2023; Parisi et al., 2022). Code generation for tool-use can be viewed as a form of semantic parsing (Shin & Durme, 2022; Krishnamurthy et al., 2017; Berant et al., 2013; Dong & Lapata, 2016) or function calling (Puri et al., 2021; Alon et al., 2019; Chen & Zhou, 2018). Our work complements this literature by providing theoretical justification (Sections 4 and 5) for the observed empirical advantages of code-based tool-use over direct natural language reasoning.

# 7. Conclusion

We introduced a three-route framework (Section 2) for disentangling representation and execution in algorithmic reasoning. The intermediate route, code generation followed by LLM simulation, lets us compare natural-language reasoning and code execution without changing both the trace representation and the executor at once. On the 40 algorithmic benchmark tasks, Route 1 and Route 2 are statistically close: Route 2 exceeds Route 1 by +0.15pp with a 95% cluster-bootstrap CI of $[-0.30, +0.61]$pp. Route 3 reaches 48.84% accuracy and exceeds Route 2 by +31.47pp with CI $[+29.20, +33.71]$pp (Section 3.2). The representation analysis in Section 4 explains the small Route 1/Route 2 gap through a shared-core nuisance model and a reconstruction intervention showing that code-derived natural-language traces retain comparable downstream utility. The execution analysis in Section 5 explains the large Route 3 gain through low recovery mass (1.61%) and high execution-win mass (33.08%, Figure 6). Taken together, these results suggest that code helps primarily by generating an executable trace that can be run reliably. In this setting, the central advantage of tool use is not the code representation, but the ability to hand the generated trace to a deterministic executor.

**Limitations.** We highlight a few limitations. First, the main experiments do not fully cover frontier reasoning models. We ran controlled-subset frontier-model ablations (Table 6) and found that the results generally hold. Second, task coverage is limited to algorithmic problems with externally verifiable answers and reliable Python execution, so open-ended, ambiguous, unsafe, or weakly specified tool use may behave differently. Third, the theory mainly builds intuition through standard decision-theoretic ideas and derives its novelty from the application, rather than providing a new general theorem about LLMs or code. Fourth, the linear models and shared-core/sufficient-statistic assumptions are motivated by practice but may fail when code omits decision-relevant information, natural language carries useful signal, or executors exploit structure outside the modeled core.

## Impact Statement

This paper presents work whose goal is to advance the field of machine learning. There are many potential societal consequences of our work, none of which we feel must be specifically highlighted here.

## Acknowledgements

This work was partially funded by ONR Contract N00014-23-1-2417. We are grateful for resources and computational support provided by the Cognitive Computation Group at the University of Pennsylvania. We also thank the reviewers for their thoughtful feedback and comments.

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

# A. Supplementary Route Results

## A.1. Complexity-Stratified Route Accuracy

We stratify the main route-accuracy evaluation by the asymptotic complexity class of each task to check whether a single complexity regime drives the aggregate route pattern.

| Complexity | Tasks | Inst. | R1 | R2 | R3 | R2–R1 | R3–R2 |
|---|---|---|---|---|---|---|---|
| $O(1)$ | 3 | 180 | 46.2 | 47.3 | 86.2 | 1.0 | 39.0 |
| $O(\log n)$ | 1 | 35 | 22.4 | 28.6 | 32.9 | 6.2 | 4.3 |
| $O(n)$ | 4 | 135 | 4.8 | 5.0 | 7.5 | 0.2 | 2.5 |
| $O(n \log n)$ | 5 | 68 | 0.0 | 0.0 | 0.0 | 0.0 | 0.0 |
| $O(n^2)$ | 16 | 297 | 10.3 | 10.5 | 38.9 | 0.1 | 28.5 |
| $O(n^2 \log n)$ | 1 | 1 | 0.0 | 0.0 | 0.0 | 0.0 | 0.0 |
| $O(n^3)$ | 4 | 37 | 0.0 | 0.0 | 0.0 | 0.0 | 0.0 |
| NP-hard | 6 | 360 | 17.6 | 16.8 | 69.7 | -0.8 | 53.0 |

*Table 1.* Route accuracy grouped by asymptotic complexity on the 40-task benchmark set in the main route evaluation. We report task and instance counts. The route differences are percentage-point gaps.

**Result.** Table 1 shows the aggregate route pattern within the larger retained complexity strata. Route 3–Route 2 is the largest observed route change, and Route 2 stays close to Route 1.

**Task mapping.** The overall task set's largest strata are $O(n^2)$ dynamic-programming, sorting, string, graph, and shortest-path tasks, plus NP-hard ILP, edge-disjoint paths, shortest-path, and TSP tasks. Smaller strata cover constant-time arithmetic, logarithmic binary search, linear selection/string matching, $O(n \log n)$ scheduling/sorting/hulls, $O(n^2 \log n)$ Kruskal MST, and cubic dynamic-programming/shortest-path routines.

## A.2. Model-Stratified Route Accuracy

We stratify the main route comparison by model to check whether the aggregate result is driven by one model family or scale regime. The Route 3 advantage persists across all six full-coverage models, while the Route 2–Route 1 difference is small and changes sign across models.

| Model | Type | Inst. | R1 | R2 | R3 | R2–R1 | $p_{2,1}$ | R3–R2 | $p_{3,2}$ |
|---|---|---|---|---|---|---|---|---|---|
| `anthropic/claude-haiku-4.5` | closed | 1113 | 23.3 | 23.6 | 47.1 | 0.3 | 0.612 | 23.5 | $2.4 \times 10^{-148}$ |
| `google/gemini-2.0-flash-001` | closed | 1113 | 17.5 | 18.1 | 51.8 | 0.7 | 0.137 | 33.7 | $8.5 \times 10^{-298}$ |
| `google/gemini-2.5-flash` | closed | 1113 | 20.7 | 20.5 | 50.2 | -0.1 | 0.823 | 29.7 | $3.1 \times 10^{-238}$ |
| `openai/gpt-4o-mini` | closed | 1113 | 14.1 | 12.9 | 49.1 | -1.2 | 0.00663 | 36.1 | $< 10^{-300}$ |
| `mistralai/codestral-2508` | open | 1113 | 15.0 | 15.8 | 52.7 | 0.8 | 0.0279 | 36.9 | $< 10^{-300}$ |
| `mistralai/mixtral-8x22b-instruct` | open | 1113 | 12.7 | 13.2 | 42.1 | 0.5 | 0.127 | 28.9 | $7.2 \times 10^{-253}$ |

*Table 2.* Route accuracy grouped by model on the 40-task analysis set. Each row contains 1,113 unique problems and 3,339 paired route evaluations; $p_{2,1}$ and $p_{3,2}$ are exact two-sided McNemar tests for Route 2 vs. Route 1 and Route 3 vs. Route 2, respectively.

**Result.** Table 2 shows that Route 3 stays above Route 2 for every full-coverage model row, so the aggregate execution gain is not driven by a single weak or strong model.

# B. Functional Similarity Checks

## B.1. Prompt Translation Shot Ablation

The functional similarity test translates code traces back into natural language as a fixed transformation. To test sensitivity to prompting, we vary the number of in-context examples used by the translator and measure translated-trace utility against native natural-language traces.

| Shots | $x$ | $x$ + native NL | $x$ + translated NL | $\Delta$ native | $\Delta$ translated | Gap |
|---|---|---|---|---|---|---|
| 0 | 32.70% | 55.70% | 42.41% | +23.00pp | +9.70pp | -13.29pp |
| 1 | 32.70% | 55.70% | 45.78% | +23.00pp | +13.08pp | -9.92pp |
| 2 | 32.70% | 55.70% | 49.58% | +23.00pp | +16.88pp | -6.12pp |
| 3 | 32.70% | 55.70% | 48.52% | +23.00pp | +15.82pp | -7.17pp |
| 4 | 32.70% | 55.70% | 49.37% | +23.00pp | +16.67pp | -6.33pp |
| 5 | 32.70% | 55.70% | 50.00% | +23.00pp | +17.30pp | -5.70pp |
| 10 (reference) | 39.00% | 56.50% | 52.00% | +17.50pp | +13.00pp | -4.50pp |

*Table 3.* Translation-additivity shot ablation for Claude Haiku 4.5. Here $x$ denotes the question-only baseline, and Gap is translated NL minus native NL. Rows 0–5 use a matched 25% subset sweep. The 10-shot row is an original main-evaluation reference and is not from the same 25% subset sweep.

**Result.** Table 3 shows that translated traces improve over the question-only baseline at every reported shot setting, but remain below native natural-language traces in every row.

## C. Additional Model Evaluations

### C.1. Recursive Language Model Evaluation

To broaden coverage beyond standard LLM forward passes, we evaluate recursive language model (RLM) execution on a 25% subset at seed 0. The RLM code and natural-language arms remain close under this executor family.

| Model | RLM Code Acc | RLM NL Acc | Better Arm |
|---|---|---|---|
| Claude Haiku 4.5 | 29.4% | 30.9% | NL |
| Gemini 2.5 Pro | 47.5% | 45.2% | Code |

*Table 4.* Recursive language model code-vs.-NL accuracy on the 25% subset, seed 0. Better Arm marks the higher row accuracy.

**Result.** Table 4 shows mixed code-vs.-NL results under RLM execution: the NL arm is higher for Claude Haiku 4.5, while the code arm is higher for Gemini 2.5 Pro. Because this is a 25% subset, we use it as model-coverage evidence rather than as the main route estimate.

### C.2. Coding-Specialized Model Evaluation

We evaluate coding-specialized models to test whether models trained on code change the Route 2–Route 1 pattern. Even for these models, replacing natural-language reasoning with simulation over code traces changes little relative to replacing simulation with actual code execution.

| Model | NL | Sim | Code Exec |
|---|---|---|---|
| `x-ai/grok-code-fast-1` (25% data) | 47.71% (167/350) | 47.71% (167/350) | 55.99% (159/284) |
| `qwen/qwen3-coder` (25% data) | 30.23% (104/344) | 26.86% (94/350) | 56.19% (168/299) |
| `codestral-2508` (original) | 19.89% (943/4740) | 23.14% (1097/4740) | 59.65% (2266/3799) |

*Table 5.* Route accuracy for coding-specialized models. NL, Sim, and Code Exec correspond to Route 1, Route 2, and Route 3; each cell reports accuracy with correct/denominator counts, and denominators are parse-normalized per arm.

**Result.** Table 5 shows that Code Exec is the highest-accuracy arm in each coding-specialized row. Grok and Qwen use 25% subset runs, while Codestral uses the original evaluation run.

### C.3. Frontier Model Controlled-Subset Evaluation

| Model | Route 1 | Route 2 | Route 3 |
|---|---|---|---|
| GPT-5.4 | 42.57% (149/350) | 41.43% (145/350) | 54.01% (175/324) |
| Claude Opus 4.6 | 52.73% (174/330) | 73.42% (116/158) | 77.54% (107/138) |

*Table 6.* Frontier controlled-subset route accuracy on the seed 1, 350-instance subset.

**Result.** Table 6 keeps Route 3 ahead in both controlled-subset rows, but the margins are smaller than in the main six-model evaluation.

# D. Failure Analysis

## D.1. Code Execution Failure Modes

Among Route 3 code-execution failures, most failures are semantic wrong answers rather than syntax, runtime, or timeout failures. This supports the interpretation that these errors often originate in the generated program specification rather than in the Python runtime itself.

| Model | Wrong answer | Syntax error | Runtime error | Time limit |
|---|---|---|---|---|
| Haiku | 56.26% | 10.07% | 1.88% | 31.80% |
| Codestral | 81.80% | 5.87% | 9.00% | 3.33% |
| Gemini 2.0 | 77.22% | 14.65% | 1.86% | 6.27% |
| Gemini 2.5 | 77.63% | 17.10% | 1.82% | 3.45% |
| GPT-4o mini | 72.93% | 4.56% | 18.97% | 3.54% |
| Mixtral | 60.67% | 4.95% | 32.14% | 2.24% |
| Total | 70.43% | 9.34% | 11.55% | 8.67% |

*Table 7.* Failure-mode distribution conditional on Route 3 code-execution failures. Rows are within-model percentages.

**Result.** Table 7 shows that wrong answers dominate among Route 3 code-execution failures. Parse-only rows, where execution completed but the returned value failed parsing or type checks, are excluded to keep the diagnostic focused on failures after execution was attempted.

