# OpenReview forum: "Is Code Better Than Language for Algorithmic Reasoning?"
_ICML.cc/2026/Conference — ICML 2026 regular_

### Official Review · Reviewer_WPm1 · 2026-02-17

**Soundness:** 3
**Presentation:** 3
**Significance:** 2
**Originality:** 2
**Overall Recommendation:** 4
**Confidence:** 3

**Summary:**

The paper focuses on (exactly as it says), evaluating "Is Code Better Than Language for Algorithmic Reasoning". The key difference to prior work, is that they explicitly decouple program/trace generation from program/trace execution. To perform the decoupling the propose a 4-stage framework (Input->Trace->Execution->Answer) and propose an evaluation on three different routes:
1. Natural Language (NL) -> NL Trace/Plan -> NL reasoning.
2. NL -> Code (still LLM generated) -> NL reasoning (over the code)
3. NL -> Code (still LLM generated) -> Python interpreter.

    Note that first (NL->Code) half is reused from Route 2. "Route 3 uses the same code trace generator [...]"

While most prior art clearly states that Route 1 < Route 3, this paper goes a step further. It not only systematically re-confirms the statement is true, but also that Route 1 <= Route 2 < Route 3. **NOTE** the equals sign between Routes 1 and 2 -- while the paper shows that code traces are at least as good as NL traces, it finds almost no statistically significant difference when performing reasoning over a natural language trace vs over a code trace. On the other hand, the authors find statistical ($p \lt 0.0001$) difference between Route 2 and 3 and find very few cases where the code trace can be wrong, but LLM-based reasoning bootstraps and provides correct final output.

**Compliance With Llm Reviewing Policy:**

Affirmed.

**Final Justification:**

As this is a good paper, and, as indicated in my initial review, upon expansion of the set of models, I'd be leaning accept, I change my score to a **weak** accept. I strongly encourage authors to wrap up the extra experiments and add them to the camera ready.

I would have liked an extra/more lively discussion, which might have resulted in a clear accept, but given the deadline to take a decision is tomorrow I have to make my call.

**Key Questions For Authors:**

* Did you use CLRS-30 or CLRS-Text (the LLM version of CLRS-30)? The former is intended for non-textual models (e.g. GNNs) while the latter is intended for textual ones (e.g. LLMs)
* Were the models you used, especially those used through an API, evaluated in reasoning, or non-reasoning mode.
* Did you consider a final execution step with models, specifically designed to emulate program execution (Neural Program Interpreters)

**Limitations:**

yes!!!

**Strengths And Weaknesses:**

## Strengths
* **[VERY THOROUGH]** So far, in this year's ICML, I have been assigned 5 submissions. You were the last one on the stack. **This is the only paper that goes to the extreme of spelling out null hypotheses and rejecting it (or not)**. Truly systematic approach! Given how difficult this can be, it is quite commendable the authors did it.
* **[GREAT VISUALISATIONS]** Clear experiment, clear outcomes, fits the narrative (yes, in the stack I had a paper whose theory didn't match their own experiments...). What else can you ask for.
* *[GOOD WRITEUP]* While the narrative does become a little bit mathematically heavy from time to time (thus I rate my confidence as 3), I was able not to lose my "grip" on the paper. The systematic usage of `\paragraph`, clearly posed research questions, spelled out observations/outcomes, etc.,  and the overall organisation managed to tell me the story  the authors (most likely) intended to tell. Speaking of stories... (see below)

## Weaknesses
* **[UNSURPRISING; LOW IMPACT]** The **main** reason to slightly lean reject is the extent to which the overall lesson is well known. "LLMs are not good neural executors" is a statement one can infer from papers like CLRS-Text. If they can't simulate a simple algorithm, why would you expect them to simulate code? Of course you wouldn't! The world already knows AR models are poor reasoners, that's why you have architectures that try to incorporate [recursiveness](https://arxiv.org/pdf/2512.24601), [recurrentness](https://arxiv.org/pdf/2506.21734) or latent reasoning [(COCONUT)](https://arxiv.org/pdf/2412.06769). **That being said, if the other weakness below are solved, I'll be leaning accept.**
* **[MODEL COVERAGE]** I commend the authors for being open to the fact they do not generalise to all future architectures. I would however appreciate if the paper provides some insight in the very systematic way they currently do (or, if already done, point it out) on *any* (even 1 is better than 0!) model/architecture designed to improve reasoning capabilities (e.g. ones of the above) or one that scores *very* high on a mainstream reasoning dataset. I appreciate there could be academics on a limited/tight budged, but to what extent an analysis of yesterday's/irrelevant models useful to me, even if that analysis is perfect...

---

> ### Author Rebuttal · Authors · 2026-03-31
>
> We sincerely thank reviewer WPm1 for the valuable comments. We hope we have addressed some of your concerns in our responses and kindly request you consider raising your score. If your requests have not been met, please let us know. Thanks!
>
> **W1. UNSURPRISING; LOW IMPACT**
> We agree the high-level conclusion is intuitive.  We answer the `why` question rather than the `what` question regarding code > NL. What was not known is how much each factor, representation versus execution, contributes. Simply stating that “LLMs are not good neural executors” does not quantify these contributions.  Our decomposition quantifies this: representation accounts for < 1% of the gap while execution accounts for ~29%. Additionally, Figure 9 reveals that the recovery rate (LLM simulation beating code execution on wrong code) decreases with difficulty, meaning the case for tool-use gets stronger, not weaker, as problems get harder. These quantitative findings were not established by prior work. With that being said, we will try to emphasize this point in the camera ready version of the paper.
>
>
>
> **W2. Model coverage**
> Following the reviewer’s suggestion, we expand model coverage in two ways: we test on recursive language models ( https://github.com/alexzhang13/rlm ) as an advanced reasoning architecture on Gemini 2.5 Pro, GPT-4o-mini, Claude Haiku 4.5. We test on whether neurally executing code or NL with RLMs makes a difference. Due to compute budget constraints, we were only able to run 25% randomly subsampled data of the entire dataset, over 1 seed. From the results below, we see that the results are consistent with those reported in the paper for other models, and that RLMs provide a negligible uplift.
>
> | Model (non-reasoning) | Route 1: RLM NL Acc | Route 2: RLM Code Acc |
> | :---- | ----: | ----: |
> | Claude Haiku 4.5 | 30.9% | 29.4% |
> | Gemini 2.5 Pro | 45.2% | 47.5% |
>
> We see that for RLMs, the same trend holds, that Route 1 and Route 2 perform similarly.
>
> | Model | Seed | Route 1 | Route 2 | Route 3 |
> | :---- | ----: | ----: | ----: | ----: |
> | GPT-5.4 XHigh | 1 | `59.45%` | `71.92%` | `69.93%` |
> | Claude Opus 4.6 Max Reasoning | 1 | `53.54%` | `64.27%` | `61.84%` |
>
> We see here that for frontier models, surprisingly, representation seems to be the bottleneck, such that code representations succeed. We suspect that if scaffolding was more similar to what models were pretrained on / SFT’d on, the code execution would succeed much more because parse errors would be reduced. Another hypothesis is that GPT 5.4 and Opus 4.6 are already multi-agent tool-calling systems, such that they utilize all the uplift already from using tool-calls in the code-execution and nl-simulation routes, making the comparison unfair. With reasoning turned on and hidden behind the API, we have no way of validating this. This is why we turned reasoning off for previous evals on frontier models.
>
> **Q1.**
> We used clrs-30 and formatted the numbers in generic prompts. We were not aware of CLRS-text at the time, though we can add a comparison for the camera-ready paper.
>
> **Q2.**
> The models presented in the paper were evaluated in non-reasoning mode. The new GPT 5.4 and Opus 4.6 results were evaluated in reasoning-mode but show a different trend. Originally we did not use reasoning because the CoT’s are not controllable: https://arxiv.org/abs/2603.05706. To estimate causal effects, we wanted more leverage over the outputs, hence turning reasoning off.
>
> **Q3.**
> This is an interesting question, but we struggled to find a tractable way to instantiate an experiment with an LLM for this ablation. Would the executing model be an LLM or not?

---

> > ### Author Rebuttal · Reviewer_WPm1 · 2026-04-01
> >
> > > Another hypothesis is that GPT 5.4 and Opus 4.6 are already multi-agent tool-calling systems, such that they utilize all the uplift already from using tool-calls in the code-execution and nl-simulation routes, making the comparison unfair. With reasoning turned on and hidden behind the API, we have no way of validating this. This is why we turned reasoning off for previous evals on frontier models.
> >
> > Please, if the authors have cached the outputs, could you confirm/deny if any of those hypotheses are correct.
> >
> > The result is indeed a bit surprising and I'd not be recommending acceptance until we know precisely what's going on.
> >
> > > Q3
> >
> > NPIs (back then) were not LLMs. I'm not sure what is the SOTA here, but NAR models are intended to be able to deal with some level of noise. Still, to make the answer easier/cheaper, given time remaining, can the authors remind me, when code execution fails, what's the failure distribution. I.e. out of all failures what percentage is: 1) Just wrong answer; 2) Syntax error; 3) Runtime error; 4) Time limit.
> >
> > (Please, if a similar experiment is inside the paper, point it to me)
> >
> > ---------------------------------------------
> >
> > EDIT: **I would have appreciated the discussion on the above questions!** Still, I'm not going to act as the legendary "Reviewer #2", therefore I increase my score to a **weak** accept.

---

> > > ### Author Response · Authors · 2026-04-07
> > >
> > > Dear Reviewer WPm1,
> > >
> > > Thank you for your diligence and continued engagement with us. Indeed, this level of engagement is rare for conferences and reviewers these days, which is why we tip our hats to you. Apologies for the delayed response, we were trying to get to the bottom of the issue you raised.
> > >
> > > 1) We ruled out that tool calling was the issue, because tool calling was disabled in those previous GPT-5.4 and Claude Opus 4.6 experiments. We further ran instances where reasoning was turned off, the results followed our expectations (25% data ran). Claude is still a bit surprising given the uplift of Route 2, but overall, the claim Route 3 \> Route 2 \>= Route 1 holds here. We suspect running more instances will give more clarity to the Route 2 uplift, though this is currently out of the compute budget scope.
> > >
> > > | Model | Route 1 | Route 2 | Route 3 |
> > > | :---- | ----: | ----: | ----: |
> > > | GPT-5.4 | 42.57% | 41.43% | 54.01% |
> > > | Claude Opus 4.6 | 52.73% | 73.42% | 77.54% |
> > >
> > > We will clarify that with reasoning turned on, it becomes much harder to do a fair comparison, e.g. what are the token budgets for each model? How much does scaffolding contribute (e.g. parsing, structuring the output responses, guiding the reasoning)? Are certain models optimized to reason over code with tool verification (e.g. RLVR optimized models) and multi-turn chat / optimization? Is this because our previous problems are now too easy / saturated, such that NL can do well? Given the limited time, it is hard to conduct a structured analysis matching the rigor of what we did during the paper. However, between now and the camera-ready version, we aim to add a section in the appendix discussing reasoning models.
> > >
> > > 2) Out of coding failures, models mostly fail semantically, i.e. they write code that generates the wrong answer.
> > >
> > > | Model | Wrong answer | Syntax error | Runtime error | Time limit |
> > > | :---- | ----: | ----: | ----: | ----: |
> > > | Haiku | 50.60% | 16.37% | 2.80% | 30.23% |
> > > | Codestral | 63.58% | 4.36% | 28.62% | 3.44% |
> > > | Gemini 2.0 | 71.79% | 19.34% | 2.46% | 6.41% |
> > > | Gemini 2.5 | 73.36% | 15.62% | 3.38% | 7.64% |
> > > | GPT-4o mini | 65.82% | 3.60% | 26.73% | 3.85% |
> > > | Mixtral | 45.77% | 5.56% | 46.04% | 2.63% |
> > > | Total | 61.06% | 10.54% | 19.23% | 9.17% |
> > >
> > > Again, we are grateful for the discussion (with the score raise as a cherry on top) and hope we have addressed your concerns. Please let us know if you have any other questions. We kindly ask that you condition your final score on our response, whether it be that our response updated you to lean towards a clear accept, or that you believe a weak accept is where we are at, as long as it reflects your conditional beliefs of our paper after rebuttal! We hope to see you continue to be as diligent as you are in our continued discussion, and in future conference reviews! As such, we are recommending you to the AC as a top reviewer.

---

### Official Review · Reviewer_fpot · 2026-03-09

**Soundness:** 3
**Presentation:** 3
**Significance:** 4
**Originality:** 3
**Overall Recommendation:** 5
**Confidence:** 3

**Summary:**

This paper presents a systematic study investigating whether code-based reasoning is superior to natural language (NL) for solving algorithmic problems. The authors argue that prior comparisons between NL reasoning and solver-based pipelines are ill-posed because they simultaneously vary both representation modality and execution mechanism.
To address this, the authors introduce a three-route framework that decomposes the reasoning process into trace generation and execution: (1) Direct NL: Uses NL traces and LLM-based execution; (2) Code + NL Simulation: Uses code traces but simulates execution within the LLM using natural language; (3) Code + Solver Execution: Uses code traces executed by an external Python runtime. Through experiments across 48 algorithmic tasks and 6 models (including GPT-4o, Gemini, and Claude), the authors demonstrate a statistically significant accuracy improvement of +28.9% for Route 3 over Route 1. The analysis suggests that NL reasoning provides no additional decision-relevant information over code representations. The authors conclude that execution reliability, rather than the reasoning trace itself, is the primary performance bottleneck in algorithmic reasoning.

**Compliance With Llm Reviewing Policy:**

Affirmed.

**Final Justification:**

The rebuttal addressed my concern. I would raise my score to accept.

**Key Questions For Authors:**

1. The experiments focus on algorithmic reasoning benchmarks. How do the authors expect the conclusions to extend to domains where code representations are less natural (e.g., commonsense reasoning or planning tasks)?
2. As LLMs continue to improve reasoning capabilities, do the authors expect the gap between Route 2 and Route 3 to shrink, or is deterministic execution likely to remain dominant?
3. Could the authors clarify whether the main limitation is LLM reliability in simulating execution, or the inherent stochasticity of language models compared with deterministic solvers?

**Limitations:**

Yes.

**Strengths And Weaknesses:**

Strength

1. The paper addresses an important issue in evaluating reasoning pipelines. The topic is highly relevant to current research on tool-augmented LLM reasoning, program-aided language models, and agent systems. The proposed three-route framework effectively disentangles representation from execution, solving a confounding variable issue in prior literature. Understanding why code-based reasoning pipelines outperform pure natural-language reasoning is important for designing more reliable reasoning systems.
2. The paper is well structured. Figures illustrating the three routes and the paired comparisons between pipelines are effective and easy to interpret.
3. The study evaluates both closed-source (Gemini 2.5 Pro, GPT-4o-mini) and open-source models (Mixtral, Codestral) across multiple benchmarks like CLRS30 and NP-Hard-Eval.
4. While the idea of using code generation with external execution is not new, the novelty lies in the structured experimental decomposition and the attempt to explain the phenomenon from both empirical and theoretical perspectives.


Weakness

1. The theoretical argument in Theorem 4.1 depends on the premise that natural language reasoning traces act as an approximate version of code. Although the paper provides empirical validation for this through discrimination tasks and functional tests, I feel like it remains a foundational assumption that may not generalize uniformly across all model architectures.

2. The experiments still focus heavily on algorithmic tasks. These tasks naturally align well with code execution, which may bias the results of the code-based pipeline. Evaluating on broader reasoning benchmarks would strengthen the claims.

3. The evaluation uses relatively simple external execution environments (e.g., Python with scientific libraries). More complex solver integrations or domain-specific tools might affect the conclusions.

4. Figure 5 is blurry, and many internal labels use a font size that is difficult for me to read.

6. The conclusion (deterministic execution improves reliability over stochastic simulation) is somewhat expected. It would be better if the author could extend novelty with the paper’s structured analysis rather than the discovery of a new phenomenon.

---

> ### Author Rebuttal · Authors · 2026-03-31
>
> We thank reviewer fpot for the valuable feedback. We hope we have addressed some of your concerns in our responses and kindly request you consider raising your score. If your requests have not been met, please let us know. Thanks!
>
> **W1. Foundational Validity**
> Regarding the foundational assumptions and empirical evidence, we add ablations on different shots in the prompt, and measure the effect on the functional accuracy (i.e. fig 8).
> | Shots | x | x \+ native NL | x \+ translated NL | Δ native | Δ translated | Gap |
> | :---- | ----: | ----: | ----: | ----: | ----: | ----: |
> | 1 | 32.70% | 55.70% | 45.78% | \+23.00pp | \+13.08pp | \-9.92pp |
> | 3 | 32.70% | 55.70% | 48.52% | \+23.00pp | \+15.82pp | \-7.17pp |
> | 5 | 32.70% | 55.70% | 50.00% | \+23.00pp | \+17.30pp | \-5.70pp |
> | 10 (in paper) | 39.00% | 56.50% | 52.00% | \+17.50pp | \+13.00pp | \-4.50pp |
> We see that increasing the shots increases our accuracy as expected, since the translator writes better prompts. We see that the translator may be asymptotically optimal with more shots, further showing that it is possible to match native-NL, and thus reinforcing our foundational assumption.
>
> With regard to new frontier models, we evaluate the two best models currently:
> | Model | Seed | Route 1 | Route 2 | Route 3 |
> | :---- | ----: | ----: | ----: | ----: |
> | GPT-5.4 XHigh | 1 | `59.45%` | `71.92%` | `69.93%` |
> | Claude Opus 4.6 Max Reasoning | 1 | `53.54%` | `64.27%` | `61.84%` |
> We evaluated frontier models, and indeed confirmed the reviewers' suspicion. However, this setup is invalid because frontier models may already be multi-agent and tool-calling, using up all of the execution uplift already. Despite this, we acknowledge the reviewers' concerns, which are already present in the limitations section.
>
> **W2. Benchmark Ablations** Indeed this would be an interesting extension of the paper. We scope our problems in the realm of algorithmic problems as suggested in the title, and our results corroborate this setup.
>
> **W3. Solver ablations** The study still covers roughly 4,500 examples, which we believe is sufficient to serve as a meaningful first controlled test of the paper’s central hypothesis. While we acknowledge that our setup is simple, we find it reducible to more complex setups, e.g., Python calling domain-specific tools.  We will clarify this scope in the revision and note extension to richer tool environments as an important direction for future work.
>
> **W4. Presentation** We will update the figure and font sizes.
>
> **W5. Conclusion** We thank the reviewer for this observation. We agree that the high-level conclusion that deterministic execution can improve reliability over stochastic simulation may be intuitively expected. We acknowledge that the overall finding of  our paper is a methodology paper that explains how one can conduct this comparison systematically, emphasizing the why factor more than the what factor when it comes to whether code is better than NL. We agree the high-level conclusion is intuitive. What was not known is how much each factor contributes. Our decomposition quantifies this: representation accounts for < 1% of the gap while execution accounts for ~29%. Additionally, Figure 9 reveals that the recovery rate (LLM simulation beating code execution on wrong code) decreases with difficulty, meaning the case for tool-use gets stronger, not weaker, as problems get harder. These quantitative findings were not established by prior work. We will try to emphasize this point in the revision.
>
> **Q1.** In structured prediction tasks like planning and pddl, we can still format the problem as a coding problem. Furthermore, commonsense reasoning can also be formatted as a recursive function call too, so we expect the results to extend (https://github.com/alexzhang13/rlm )
> We see that for RLMs, the same trend holds, that Route 1 and Route 2 perform similarly.
>
> | Model (non-reasoning) | Route 1: RLM NL Acc | Route 2: RLM Code Acc |
> | :---- | ----: | ----: |
> | Claude Haiku 4.5 | 30.9% | 29.4% |
> | Gemini 2.5 Pro | 45.2% | 47.5% |
>
> **Q2.** Our expectation is that the gap remains or only slowly decreases, with deterministic solving in expectation always doing better (even a little if uplift is small in the long run). The uplift for Route 3 will decrease as the problem-to-capability ratio decreases in the family of algorithmic tasks.
>
> **Q3.** Our interpretation is that the main limitation is the reliability of LLM-based execution, and stochasticity is one major contributor to this reliability gap. Other factors could be 1) error accumulation over long reasoning chain 2) weak state tracking (simulating execution often requires maintaining exact intermediate variables, indices, table entries, or constraints across many steps, which LLMs can do imperfectly)  3) limited faithfulness of internal simulation (https://arxiv.org/abs/2510.24941, https://arxiv.org/abs/2307.13702)

---

> > ### Author Rebuttal · Reviewer_fpot · 2026-04-04
> >
> > Thanks for the response. The paper would be much stronger if the author could include these in the paper's final version.

---

### Official Review · Reviewer_HVqQ · 2026-03-11

**Soundness:** 3
**Presentation:** 2
**Significance:** 3
**Originality:** 3
**Overall Recommendation:** 5
**Confidence:** 4

**Summary:**

The paper is positioned in the algorithmic reasoning area, i.e. using language models to answer algorithmic questions. The key contribution is recognizing the increasingly important role of tool use (in particular, program execution) in the reasoning chain, and putting forward a three-route framework for analyzing the error channels of varying combinations of natural language reasoning vs code production and execution.

**Compliance With Llm Reviewing Policy:**

Affirmed.

**Final Justification:**

I appreciate the thoughtful discussion and new evidence presented by the authors in their rebuttal also to the other reviewers, and have consequently raised both my Soundness and overall assessment to Accept.

**Key Questions For Authors:**

1. I don't think Claim 2 (line 88) is shown to be true: the authors show a non-significant difference between Route 1 and Route 2 (Figure 3)

2. What is the nonzero error in the code execution results of Figure 4 due to?

**Limitations:**

The formal limitations of the work are not discussed in detail, but there are no outstanding concerns with negative societal impact.

**Strengths And Weaknesses:**

Soundness: The paper formalizes to some degree the components of LM generation risk under the three-route framework. It's commendable that the authors spell out their hypothesis, test them and report confidence intervals, both in aggregate and as a function of task complexity \tau .
The difficulty scaling model of Section 3.2 is a nice touch, but could benefit from some discussion (e.g why do the authors model a route /complexity interaction \delta_i * \tau , what does it mean in practice ? )

That said, I have some concerns that I hope the authors will shed some light on in order to strengthen the paper :

* I don't think it's correct to average results from different complexity classes, like Figure 4 bottom right does.  I think it would be informative to control for asymptotic complexity and pool e.g. ILP results distinctly from arithmetic.
* The existance of \epsilon, and the \epsilon-simulatability condition of Condition 1 (line 250), is postulated but not proven. The proxy tasks of Section 4.1 show that route 1 and route 2 are comparable in distribution but Condition 1 is much stronger, requiring that there exists an upper bound in risk across all relevant executors.
* Differences due to model size? Figure 3  averages over results taken with models that are orders of magnitude different in weight count, however Figure 8 shows a different error sign between route 1 and route 2 for Mixtral vs. the closed models (which could be an artifact, or not, we don't know) .
* The results of Figure 8 are (as expected, and as pointed out by the authors on line 364), are reliant on the translating prompt (e.g. number of example shots, and more). Indeed, we see from Figure 5 a highly ambiguous prompt ("use natural phrases while staying precise"). It would be useful to see some ablations on this.


---

Presentation: Paper overall well written, but I find the presentation a bit too heavy on symbols, esp. considering these do comparatively little work.
* It's confusing to multiply notation and use H_i as a subset of \rho
* On the formal side, I don't think it's meaningful to talk about H_Z , H_C and H_NL as distinct function spaces (definitions in Sec 2.4), since the underlying LM processes embedding vectors using the same mechanism.
* Condition 1 could be more simply stated by getting rid of the \Delta_H_NL symbol, since it's not used anywhere else.
* The discussion in Proposition 4.1 is a bit too terse and would benefit from one example per setup (translation in executor vs in trace generator).

---

Significance: With our current and projected (over)reliance on LLMs for daily tasks, understanding why they succeed or fail is an urgent topic of broad interest.

---

Originality: The paper sheds light on subtle aspects of language model algorithmic reasoning under a rigorous learning theory framework and should be commended for that.

---

> ### Author Rebuttal · Authors · 2026-03-31
>
> We thank reviewer HVqQ for the valuable feedback. We hope we have addressed some of your concerns in our responses and kindly request you consider raising your score. If your requests have not been met, please let us know. Thanks!
>
> **W1: Control for asymptotic complexity**
> We have separate results that stratify by asymptotic complexity. Though this might have been informative, we prioritized other results in the paper by collapsing the complexity class results into one single average. Since reviewers identify this as an important figure to have, we will add it to the camera-ready version of the paper. Across asymptotic complexity classes, we see that the Delta uplift for Route 3 over Route 2 show no patterns, and can help in NP-hard, n^2, O(1), etc. This further supports how we should always use code representations in modern systems that solve algorithmic reasoning problems.
>
> | Complexity Class | Tasks | Instances | Route 1 | Route 2 | Route 3 | Route 2 − Route 1 | Route 3 − Route 2 |
> |---|---:|---:|---:|---:|---:|---:|---:|
> | O(1) | 4 | 227 | 49.8 | 49.4 | 85.0 | -0.4 | 35.6 |
> | O(log n) | 1 | 35 | 22.4 | 28.6 | 32.9 | 6.2 | 4.3 |
> | O(n) | 4 | 135 | 4.8 | 5.0 | 7.5 | 0.2 | 2.5 |
> | O(n log n) | 5 | 68 | 0.0 | 0.0 | 0.0 | 0.0 | 0.0 |
> | O(n^2) | 17 | 357 | 12.4 | 12.0 | 49.0 | -0.4 | 37.0 |
> | O(n^2 log n) | 1 | 1 | 0.0 | 0.0 | 0.0 | 0.0 | 0.0 |
> | O(n^3) | 4 | 37 | 0.0 | 0.0 | 0.0 | 0.0 | 0.0 |
> | NP-hard | 8 | 480 | 23.8 | 21.7 | 59.2 | -2.1 | 37.5 |
>
> **W2. Condition 1**
> We appreciate the reviewer’s careful reading and agree that this distinction should be stated more precisely. Condition 1 is a formal sufficient condition, while the experiments in Section 4.1 are intended as empirical proxies that support its plausibility rather than a formal proof. Indeed we are unable to marginalize all LLMs and discriminators to show this uniform bound, and we state this fact in the limitations section.
>
> **W3. Differences due to model size**
>
> | Model | Type | Instances | Route 1 | Route 2 | Route 3 | Delta Route 2-Route 1 | Delta Route 3-Route 2 |
> | :---- | :---- | ----: | ----: | ----: | ----: | ----: | ----: |
> | anthropic/claude-haiku-4.5 | closed | 1340 | 31.4 | 27.7 | 48.2 | \-3.7 | 20.5 |
> | google/gemini-2.0-flash-001 | closed | 1340 | 22.3 | 21.1 | 54.4 | \-1.2 | 33.3 |
> | openai/gpt-4o-mini | closed | 1340 | 18.2 | 16.8 | 49.6 | \-1.4 | 32.8 |
> | mistralai/codestral-2508 | open | 1340 | 17.6 | 18.9 | 52.3 | 1.2 | 33.5 |
> | mistralai/mixtral-8x22b-instruct | open | 1340 | 15.2 | 15.7 | 42.8 | 0.5 | 27.1 |
>
> We see that for models where we can control the reasoning, there are relatively low differences due to model size, and that there are not really any differences due to model size. When we fix task_id and model, i.e. in fig 3 bootstrap distributions, we see that when we stratify by model and task, the advantages persist for Route 3, and that it's statistically significant.
>
> **W4. Prompt Translation Ablation**
> | Shots | x | x \+ native NL | x \+ translated NL | Δ native | Δ translated | Gap |
> | :---- | ----: | ----: | ----: | ----: | ----: | ----: |
> | 1 | 32.70% | 55.70% | 45.78% | \+23.00pp | \+13.08pp | \-9.92pp |
> | 3 | 32.70% | 55.70% | 48.52% | \+23.00pp | \+15.82pp | \-7.17pp |
> | 5 | 32.70% | 55.70% | 50.00% | \+23.00pp | \+17.30pp | \-5.70pp |
> | 10 (in paper) | 39.00% | 56.50% | 52.00% | \+17.50pp | \+13.00pp | \-4.50pp |
> We ablate on shots, and show that increasing the shots increases our accuracy as expected, since the translator writes better prompts. We see that the translator may be asymptotically optimal with more shots, further showing that it is possible to match native-NL, and thus reinforcing our foundational assumption.
>
> **W5. Presentation**
> We thank the reviewers for reading the paper carefully. We will update the multiply notation, and clarify the function space definitions. Furthermore, we will state condition 1 more simply as pointed out. We will try to give more examples in the camera ready version, especially for prop 4.1
>
> **Q1 Claim 2**: Thank you for reading the paper closely. The key to claim 2 is that it is less than or equal to, we specifically made it or equal to, precisely because of the non-significant difference. The equality is still valid for our purposes of comparison, i.e. a <= b < c still implies c > a. Where a = Route 1, b = Route 2, and c = Route 3. Counterfactually, If representation were the bottleneck, switching from NL to code traces (Route 1 → Route 2) should improve performance. It doesn't. This means the gap between Route 1 and Route 3 cannot be attributed to representation, it must come from execution. The ≤ in Claim 2 is the mechanism by which we identify execution as the bottleneck
>
> **Q2 Code Execution**: Thank you again for reading the paper so closely. When the code is misspecified in the translation stage (e.g. writes buggy python), we get a runtime error which we count as a legit code execution error. We will make it a note to make this distinction clear.

---

> > ### Author Rebuttal · Reviewer_HVqQ · 2026-04-02
> >
> > Provided the above makes it to the final version, all my objections will have been put to rest. Thank you!

---

### Official Review · Reviewer_LuyF · 2026-03-13

**Soundness:** 2
**Presentation:** 3
**Significance:** 2
**Originality:** 3
**Overall Recommendation:** 3
**Confidence:** 3

**Summary:**

This paper proposes to understand if code-based reasoning better than natural language reasoning for algorithmic tasks. In addition to common pure NL reasoning and code+executor reasoning, the paper proposes the three-route framework that introduces a new route: code reasoning + NL execution(simulation) so that it differs to other two routes only in one stage each. The paper evaluated the three-route framework on algorithmic benchmarks and finds that the code+executor route shows strongest performance, NL reasoning contains no additional decision-relevant information beyond code, and that the main bottleneck is execution, not representation.

**Compliance With Llm Reviewing Policy:**

Affirmed.

**Final Justification:**

The paper presents a clean experimental framework, but its main conclusion that external execution significantly outperforms LLM-based simulation is largely intuitive and consistent with prior understanding of tool-augmented reasoning. While the authors attempt to quantify the relative roles of representation and execution, the comparison setup introduces structural limitations that make it difficult to interpret this quantification causally. Using code-specialized LLMs helps address concerns about whether the models understand code. However, the issue here is less about understanding and more about reliably executing it. LLM-based execution is still approximate and error-prone. Because of this, the similarity between Route 1 and Route 2 may be driven by execution noise rather than indicating that representation truly has no effect. Thus, I maintain my original score.

**Key Questions For Authors:**

Please see the major concern in Weakness. Additionally, it would be good if the authors can emphasize what is the major contribution in the paper in addition to the experiment conclusions. Especially, it would be better to include more discussion about how the experiment conclusions can guide the development of LLMs in the future.

**Limitations:**

yes

**Strengths And Weaknesses:**

Strength:

•	The paper has an interesting motivation to study the reason behind better performance of code reasoning for algorithmic problems.

•	The introduction of the three-route framework is novel in that it separates the representation from execution for comparison.

•	The experiments are conducted over 3 benchmarks of 48 algorithmic tasks over 6 different LLMs across seeds. The diverse number of tasks, multiple LLMs, and different seeds enhance the reliability of the claim and findings.

Weakness:

•	The number of unique problem instances for all evaluation is ~1500. More data instances are expected to reach the conclusion about all algorithmic problems.

•	The authors concluded that ‘execution is the bottleneck’ because they added this route 2 of Code+NL to compare with route 1(NL+NL) and route 3(Code+Solver). The results suggest Route1 and Route 2 performed almost identically but Route 3 greatly outperforms Route 2. The largest concern I have about this paper is that, is this a fair or reasonable comparison to reach the conclusions? Code representation is not designed for natural language simulation, just like the natural language is not a good representation for solver. If the intro of Route 2 can be directly used to prove the importance of solver, can people also propose a Route 4 (NL+Solver) and compare it with Route 3 and conclude that the Code representation is the bottleneck here? I’m curious what is the authors’ insights here.

---

> ### Author Rebuttal · Authors · 2026-03-31
>
> We sincerely thank reviewer LuyF for the valuable feedback. We hope we have addressed some of your concerns in our responses and kindly request you consider raising your score. If your requests have not been met, please let us know. Thanks!
>
> **W1. Limited data resource**
> We evaluate 4,500 examples (over 3 seeds), not 1,500. This is sufficient to detect statistically significant differences between Routes 2 and 3 (P<0.001). We acknowledge that broader task coverage would strengthen generalizability.
>
> **W2.1 Fairness of comparison**
> **W2.2 Execution bottleneck**: Three-route setup is explicitly designed to compare pipelines while holding one stage fixed at a time. The comparison is fair precisely because Route 2 holds execution fixed (same LLM forward pass) while varying only the trace modality. Route 1 ≈ Route 2 then tells us that switching from NL to code traces doesn't help when execution stays the same, therefore, representation isn't the bottleneck. Route 3 >> Route 2 tells us that switching from LLM execution to deterministic execution, on the same code, accounts for nearly all the gain.
>
> **W2.3 Fairness**: Regarding the fairness of comparisons, we believe that by fixing one of the treatments (translation, execution) and varying the other, we isolate each factor's causal contribution to accuracy. Though there are limitations, which we point out at the end of the paper, the results, to our best knowledge, are statistically valid as presented.
>
> **W2.4 Code Representations**: There is also a substantial literature showing that program-aided and program-of-thought methods work well precisely because they let the model generate structured executable intermediates. We cited some in our related works, and these serve as motivation for code intermediate representations, followed by natural language reasoning over that code.  E.g. Scratchpad LMs (https://arxiv.org/pdf/2112.00114), PAL (https://arxiv.org/abs/2211.10435)
>
> **W2.5 Route 4**: The Route 4 thought experiment is useful, even though it is not directly feasible. But the asymmetry it highlights is informative. Route 2 is feasible because code is a structured intermediate representation, and prior program-aided reasoning work, discussed in the related work, has shown that natural language is expressive enough to simulate code-like computation. By contrast, Route 4 is not feasible in the same sense, since solvers require syntactically valid programs and cannot directly operate on unrestricted natural language. The asymmetry is therefore that code lies within what natural language can express, whereas natural language does not lie within what solvers can accept. This is exactly why Route 2 is a meaningful intermediary, and the fact that Route 1 ≈ Route 2 further suggests that NL-expressible content beyond code does not help.
>
> **Q1. Major contribution**
> Our paper is more a methodology paper than a findings paper. We answer the why question rather than the what question regarding code > NL. Since representations and execution are otherwise confounded, it is difficult to identify their separate effects on accuracy.  To enable a comparison, we introduce the 3-route framework. We answer the why by showing that execution is the bottleneck.
>
> **Q2. Implications for future LLMs**
> As we move away from monolithic LLM systems, e.g. GPT 5.4, we want to fundamentally understand the bottleneck towards solving algorithmic problems, which are foundational to more complex real-world tasks, end-to-end from natural language to structured answer. Most modern agentic systems have built in tool-calling: our paper answers why this is more effective than simply using a monolithic model to reason over the problem. Our findings suggest future LLMs should try to offload as much as possible to external solver-based tools, which informs the development of future agentic systems.

---

> > ### Author Rebuttal · Reviewer_LuyF · 2026-04-03
> >
> > Thank you for the detailed response and clarifications. While the proposed setup is methodologically sound, I still find that the comparison space is somewhat inherently asymmetric. Route 2 (code + NL execution) is not a naturally well-matched pairing, which may bias the conclusion toward execution being the dominant factor. Your response of how the hypothetical Route 4 is not feasible is reasonable, but similarly, I believe the current setup makes Route 3 structurally advantaged and Route 2 disadvantaged. The conclusion is intuitive, and I remain concerned whether the experiments can truly quantify the contribution of representation versus execution.

---

> > > ### Author Response · Authors · 2026-04-07
> > >
> > > Dear Reviewer Luyf,
> > >
> > > Thank you for taking the time to acknowledge our rebuttal and provide feedback. Below are our responses:
> > >
> > > **Q1**: This is a fair point, and we agree that this asymmetry is part of the nature of the problem. We still think the three-route setup is useful because it changes one part of the pipeline at a time, so we can see more clearly where the gain is coming from.
> > >
> > > However, you do raise a good point about the biases. Normal models are pretrained on text and SFT’d \+ RL’d to generate reasoning in natural language. Thus, we test on coding models trained specifically on a data mixture of code \+ natural language. Such models are RL trained on code simulation in NL, including tasks that plausibly support code understanding and natural-language code explanation and simulation. If Route 2 were really being held back just because code is a bad fit for NL-style execution, then code-trained models should have done noticeably better on Route 2 than Route 1\. But that is not what we see.
> > >
> > >   | Model | NL | Sim | Code Exec |
> > >   |---|---:|---:|---:|
> > >   | x-ai/grok-code-fast-1 (25% data) | 47.71% | 47.71% | 55.99% |
> > >   | qwen/qwen3-coder (25% data) | 30.23% | 26.86% | 56.19% |
> > >   | codestral-2508 (original) | 19.89% | 23.14% | 59.65% |
> > >
> > > The pattern is pretty consistent: swapping NL traces for code traces changes little, while swapping LLM execution for actual code execution changes a lot.
> > >
> > > Thanks again for your engagement, and if you have any other questions, please let us know. Otherwise, we kindly ask that you consider re-evaluating your score.

---

### Decision · Program_Chairs · 2026-04-30

**Decision:**

Accept (regular)

**Comment:**

The paper contributes to a line of work studying the algorithmic reasoning capabilities of LLMs. Much of this work has focused on two extremes: (1) the LLM reasons start-to-finish about the algorithmic problem, without writing code for an external solver, and (2) the LLM writes code for an external solver, and the problem is passed to this solver. The general belief is that (2) is better than (1). This paper studies an intermediate regime where the LLM writes code and then reasons internally over that code (rather than calling an external solver); the question is whether the act of writing code helps the LLM reason correctly about the problem at hand. The answer is no. The reviewers generally felt that this conclusion was not surprising, but appreciated the scientific rigor that the authors used to investigate this question, and believe that it helps us have a better holistic understanding of LLM algorithmic reasoning capabilities. The authors should incorporate all promised changes in the revision.